# Recovery-induced tipping in Stommel's kicked ocean box model

Peter Kelly☺, Xinyi Leng☺, Pascal Cogan, Albert Jing, Katherine Meyer ‡*

Department of Mathematics and Statistics, Carleton College, Northfield, Minnesota, United States of America

☺ These authors contributed equally to this work.
‡ Senior author
* kjmeyer@carleton.edu

## Abstract

Ocean circulation, a key driver of global climate, is subject to recurrent disturbances including freshwater input from glacial melt. Stommel's idealized two-box model predicts two distinct stable regimes of thermohaline ocean circulation and offers a conceptual framework in which to explore the effects of salinity perturbations. A study that superimposed discrete salinity kicks onto the flow of the otherwise continuous model revealed a counterintuitive phenomenon: extending the recovery period between salinity kicks caused a trajectory to tip into an alternative basin of attraction. Here we analyze the recovery-induced tipping phenomenon in Stommel's kicked model across a broader swath of disturbance parameters, finding that the phenomenon is robust across a significant set of kick sizes. Furthermore, bifurcation analysis of flow-kick fixed points reveals that recovery-induced tipping stems from an alternative attractor migrating across an original separatrix as flow times vary. This migration also occurs in a continuous analog of salinity perturbation. Although the recovery times associated with seasonal or decadal freshwater influxes far exceed those associated with recovery-induced tipping, the qualitative dynamics we uncover in Stommel's perturbed model alert us that similar phenomena may occur in other models and real-world systems.

## 1 Introduction

Stommel's two-box model of thermohaline oceanic circulation [1] laid a theoretical foundation for scientists to study tipping between alternative stable regimes in the Atlantic Meridional Overturning Circulation (AMOC) [2], a climate subsystem that transfers heat and salt from low to high latitudes. In this paper we incorporate pulses of freshwater into Stommel's model as a case study for tipping via non-intuitive flow-kick dynamics.

Flow-kick disturbance models are a framework for impulsive differential equations that uses maps to analyze the interplay between repeated, discrete perturbations and

**Data availability statement:** The MATLAB scripts and functions used in the study are available in the public GitHub repository at https://github.com/katejmeyer/KickedStommel.

**Funding:** KM was supported by Grant #2418973 from the National Science Foundation (https://www.nsf.gov). The National Science Foundation did not play any role in the study design, data collection and analysis, decision to publish, or preparation of the manuscript.

**Competing interests:** The authors have declared that no competing interests exist.

continuous recovery processes. This interplay arises within a variety of applications including viral exposure [3], pest management [4], episodic herbivory [5], wildfires in savannas [6], rainfall in drylands [7], and hurricanes in coral reefs [8]. Across applications, a flow-kick model composes the time-$t$ map of a flow, which represents continuous recovery, with a "kick" map representing perturbation. The result is a flow-kick map (also termed "hybrid map" [9]) that can provide insight into disturbance dynamics. For example, fixed points of the flow-kick map represent balance between disturbance and recovery [10], while bifurcations in parameters tied to recovery time or kick magnitude represent thresholds in disturbance outcomes [11].

Flow-kick models exhibit heightened complexity across multiple comparison points. First, interspersing flows and kicks can generate complex dynamics absent from either process alone. Numerical studies of a kicked van der Pol system reveal a strange attractor, despite the flow being planar and the kick being a simple translation [9]. More generally, analytic studies of kicked limit cycles have identified conditions that produce horseshoes and chaos [12]. Second, flow-kick maps can exhibit novel dynamics relative to continuous models of disturbance. For example, modeling fires as discrete kicks to biomass in a tree-grass system revealed a transcritical bifurcation that was not present in an analogous continuous model of fires [6].

Here we examine an unexpected behavior reported by [10] that occurs when Stommel's two-box model of ocean circulation [1] is interrupted periodically by kicks to the salinity variable. Stommel's planar system, described in detail in Sect 2, has two stable equilibria: $A$, representing a temperature-dominated circulation pattern akin to the Atlantic Meridional Overturning Circulation, and $C$, representing a salinity-dominated circulation pattern. The essential surprise reported by [10] is that lengthening the recovery period (flow) between kicks causes a trajectory originating in $A$'s basin of attraction to tip into the basin of attraction of $C$ (see Fig 3b, 3c). For brevity, we refer to this as "recovery-induced tipping."

Tipping in dynamical systems involves a change in the long-term fate of a trajectory towards a different attractor. Processes known to trigger tipping in climate models include bifurcations, noise, and quickly changing parameters (rate-induced tipping) (see [13] and references therein). To this list we add recovery-induced tipping, both in the forward direction—in which lengthening the recovery period between disturbances can trigger a shift to an alternative attractor—and in the backward direction—in which increasing disturbance frequency can trigger a shift back. This latter possibility, discussed in detail in Sect 6.1, may be relevant in Earth systems as climate change increases the frequencies of disturbances such as hurricanes and wildfires.

To clarify and contextualize recovery-induced tipping in Stommel's kicked model, we address the following questions:

1. How robust is recovery-induced tipping within Stommel's kicked model?
2. What mechanisms underlie the recovery-induced tipping phenomenon in Stommel's kicked model?
3. Can a continuous model of salinity disturbance generate behavior analogous to recovery-induced tipping?

In Sect 3 we explore basin outcomes over a broader swath of $(\tau, \kappa)$ parameter space and find that recovery-induced tipping is robust for kick sizes between 0 and about 0.6. Bifurcation analysis in Sect 4 reveals the mechanism underlying the phenomenon: as flow times decrease, a globally stable fixed point migrates from the basin of $C$ to the basin of $A$. We show in Sect 5 that the same fundamental mechanism also occurs in a continuous analog to the Stommel flow-kick model. We conclude in Sect 6 by discussing the implications of recovery-induced tipping in dynamic models of resilience, comparing the kicks and flows analyzed in Stommel's model to geophysical measurements, and identifying directions for future study.

## 2 Model

### 2.1 Stommel's two-box model of ocean circulation

Stommel's classic conceptual model of ocean circulation [1] contains two connected boxes of water (Fig 1): one at low latitudes with temperature $T_1$ and salinity $S_1$ and another at high latitudes with temperature $T_2$ and salinity $S_2$. Warmer surroundings and net evaporation at low latitudes are represented as an external bath with relaxation temperatures $T_1^*$ and salinity $S_1^*$. Similarly, an external bath around the high-latitude box with relaxation temperature $T_2^* = -T_1^*$ and salinity $S_2^* = -S_1^*$ reflects the cooler surroundings and net precipitation at high latitudes. One can interpret the parameters $T_{1,2}^*$ and $S_{1,2}^*$ as the temperatures and salinities that the water in each respective box would equilibrate to in the absence of circulation between the boxes.

The circulation $q$ through the bottom capillary tube and surface channel is driven by density differences between the fluids in the boxes, which are in turn determined by temperature and salinity (see Eq 2). Importantly, the model assumes instantaneous mixing within each box, so does not resolve spatial variation within the boxes such as vertical stratification. The model also ignores inertia associated with circulation.

Coupling linear relaxation of the low- and high-latitude boxes towards respective temperatures $T_1^*$, $T_2^*$ and salinities $S_1^*$, $S_2^*$ together with the circulation $q$ that mixes fluid between the boxes yields the system of ODEs

$$\frac{dS_1}{dt} = d(S_1^* - S_1) + |q|(S_2 - S_1) \tag{1a}$$

$$\frac{dS_2}{dt} = d(S_2^* - S_2) + |q|(S_1 - S_2), \tag{1b}$$

$$\frac{dT_1}{dt} = c(T_1^* - T_1) + |q|(T_2 - T_1) \tag{1c}$$

$$\frac{dT_2}{dt} = c(T_2^* - T_2) + |q|(T_1 - T_2) \tag{1d}$$

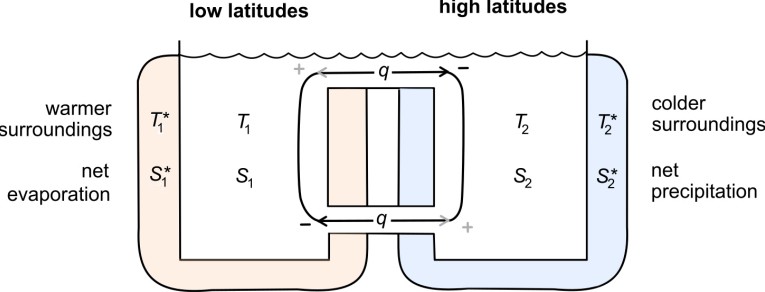

**Fig 1**. **Stommel's two-box model.** A low-latitude box of ocean water with temperature $T_1$ and salinity $S_1$ is connected via a capillary tube and surface channel to a high-latitude box with temperature $T_2$ and salinity $S_2$. Baths with temperatures $T_1^*$, $T_2^*$ and salinities $S_1^*$, $S_2^*$ represent surrounding conditions at low and high latitudes.

in which $c > 0$ and $d > 0$ parameterize the temperature and salinity relaxation rates, and the circulation rate

$$q = k(\alpha(T_2 - T_1) - \beta(S_2 - S_1)) \tag{2}$$

is a function of the temperature and salinity states that determine density differences between the boxes [1,14]. The proportionality constants $k$, $\alpha$, and $\beta$ are positive. Cooler temperatures at high latitude ($T_2 < T_1$) promote higher fluid density there and can make the circulation rate $q$ negative. In this temperature-driven scenario, fluid sinks in the high-latitude box, passes through the bottom capillary, and mixes with the low-latitude box before moving along the surface back to the high-latitude box. On the other hand, lower salinities at high latitudes ($S_2 < S_1$) decrease fluid density there and promote a positive circulation $q$ in the opposite direction. The model's bistability with respect to these two circulation patterns, discussed below, has attracted particular attention from oceanographers and climate scientists in the context of the Atlantic Meridional Overturning Circulation [15,16].

By imposing the symmetries $T_2^* = -T_1^*$, $S_2^* = -S_1^*$ and nondimensionalizing, one can reduce Eqs (1) and (2) to the planar system

$$x' = \delta(1 - x) - \frac{1}{\lambda}|Rx - y|\,x \tag{3a}$$

$$y' = 1 - y - \frac{1}{\lambda}|Rx - y|\,y \tag{3b}$$

(see S1 File). In Eqs (3), $x = (S_1 - S_2)/(S_1^* - S_2^*)$ is a normalized salinity difference (anomaly) between the two boxes, $y = (T_1 - T_2)/(T_1^* - T_2^*)$ is a normalized temperature anomaly, and $\delta = d/c$ gives the ratio of the salinity and temperature relaxation rates. The additional dimensionless parameters are $R = \beta(S_1^* - S_2^*)/\alpha(T_1^* - T_2^*)$ and $\lambda = c/2\alpha k(T_1^* - T_2^*)$.

The phase portrait in Fig 2 illustrates bistability of the system (3) with Stommel's parameter choices $\delta = 1/6$, $R = 2$, and $\lambda = 1/5$. Equilibria and flow trajectories were plotted in MATLAB R2024a using the script `Fig2.m`. Two stable equilibria—the nodal sink $A$ and spiral sink $C$—are separated by the saddle equilibrium $B$. Each point $(x,y)$ in state space determines a nondimensional circulation rate $\frac{1}{\lambda}|Rx - y|$, which is positive below the line $y = 2x$ (hatched pattern) and negative above $y = 2x$ (no pattern). The equilibrium $A$ lies in the region of negative circulation associated with sinking fluid at high latitudes, while the equilibrium $C$ lies in the region of positive circulation. When we analyze the long-term fates of trajectories under various salinity perturbations below, we pay particular attention to both the direction of circulation (positive or negative) and the basin of attraction (shaded blue for $A$ and light yellow for $C$). Combinations of these two attributes define four regions labeled (i)–(iv) in Fig 2.

## 2.2 Salinity disturbances via kicks and flows

We use the flow-kick framework to explore how periodic freshwater inputs affect ocean circulation in Stommel's model. In this framework, the ODE system given by Eqs (3) generates the flow function $\vec{\varphi}_\tau(x, y)$ that models unperturbed ocean dynamics. To represent salinity kicks $\kappa$ that occur every $\tau$ time units, we iterate the flow-kick map

$$\vec{G}_{\tau,\kappa}\begin{pmatrix} x \\ y \end{pmatrix} = \vec{\varphi}_\tau\begin{pmatrix} x \\ y \end{pmatrix} + \begin{bmatrix} \kappa \\ 0 \end{bmatrix}, \tag{4}$$

which composes the time-$\tau$ map of the flow with translation by $\kappa$ in the salinity variable $x$. Because $x = (S_1 - S_2)/(S_1^* - S_2^*)$, a positive kick $\kappa$ could represent a decrease in high-latitude salinity $S_2$, an increase in low-latitude salinity $S_1$, or both. In this paper we will focus on the former interpretation, attributing salinity kicks to pulses of freshwater input at high latitudes.

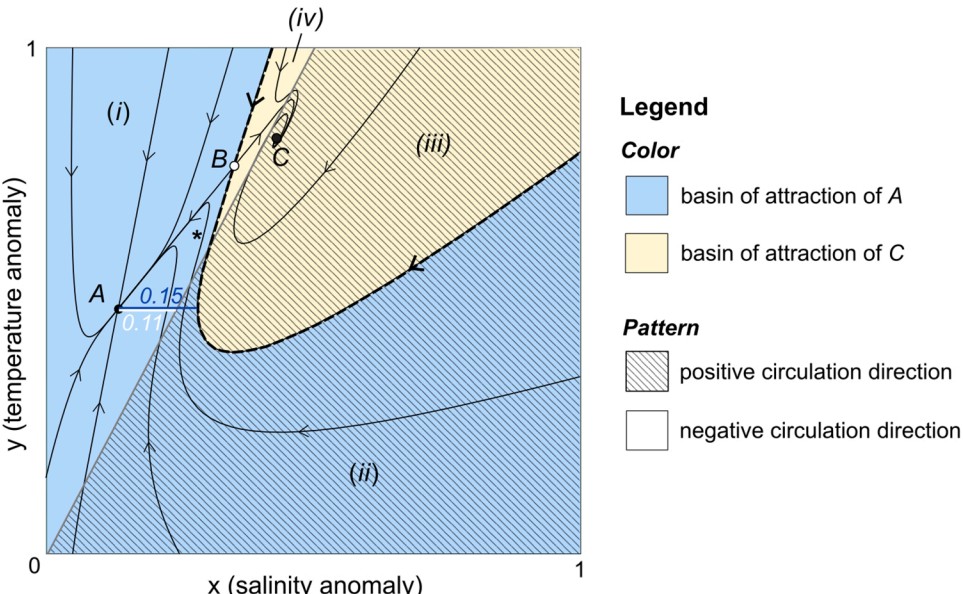

**Fig 2**. **Stommel phase portrait.** Solution trajectories for Eq (3) with $\delta = 1/6$, $R = 2$, and $\lambda = 1/5$ are plotted as thin black lines. The system has three equilibria: the stable node $A$, saddle $B$, and stable spiral $C$. The stable manifold of $B$ (heavy dashed line) is the separatrix between the basin of $A$ (blue) and the basin of $C$ (light yellow). The hatched pattern indicates the region of positive circulation and no pattern indicates negative circulation. Combinations of basin and circulation direction partition state space into four regions labeled (i)-(iv).

Kicks are a stylized model of recurrent salinity perturbations in the real ocean. For example, Greenland contributes annual pulses of freshwater flux to the North Atlantic in the summer months, in addition to baseline flow [17]. Kicks represent the simplifying assumption that all water associated with a pulse arrives instantaneously in time. See Sect 6.2 for further comparison between the kicked ocean box model and geophysical processes.

## 3 Basin outcomes and circulation reversal

Meyer and colleagues [10] reported the non-intuitive result that increasing the recovery period $\tau$ between salinity disturbances $\kappa$ could trigger tipping out of the basin of $A$ in Stommel's model. In particular, a flow-kick trajectory starting from the equilibrium $A$ with $|\kappa| = 0.1$ stabilized in the basin of $A$ for recovery time $\tau = 0.1$ (Fig 3b), but for a longer recovery time $\tau = 1$ the trajectory stabilized in the alternate basin of $C$ (Fig 3c).

In Fig 3a we contextualize this recovery-induced tipping within $(\tau, \kappa)$ parameter space by mapping whether a flow-kick trajectory from $A$ stabilizes in the basin of $A$ (blue) or $C$ (light yellow) for $0 < \tau \leq 10$ and $0 < \kappa \leq 1$. Flow-kick trajectories were simulated numerically in MATLAB R2024a using the script `Fig3a.m`. Using a grid of $(\tau, \kappa)$ values with a spacing of 0.05 for $\tau$ and 0.005 for $\kappa$, we tested the basin fate of each flow-kick trajectory by ceasing kicks after 100 iterations of the flow-kick map and tracking whether the trajectory flowed to within 0.01 of $A$ or $C$.

The points labeled **b** and **c** correspond to the $(\tau, \kappa)$ combinations presented in [10] and illustrated in Fig 3b, 3c. By looking from left to right across the line with slope 0.421 in Fig 3a, we see that increasing $\tau$ can change a trajectory's long-term basin fate from $A$ to $C$ not only for $\kappa = 0.1$ but also for positive $\kappa$ less than about 0.6. In this sense the recovery-induced tipping phenomenon reported by [10] is fairly robust within the model. Sect 4 analyzes the mechanism underlying this recovery-induced tipping: a saddle-node bifurcation coupled with a stable flow-kick fixed point migrating across the separatrix between the basins of $A$ and $C$. (See Sect 5 Continuous salinity disturbances for the significance of the slope 0.421.)

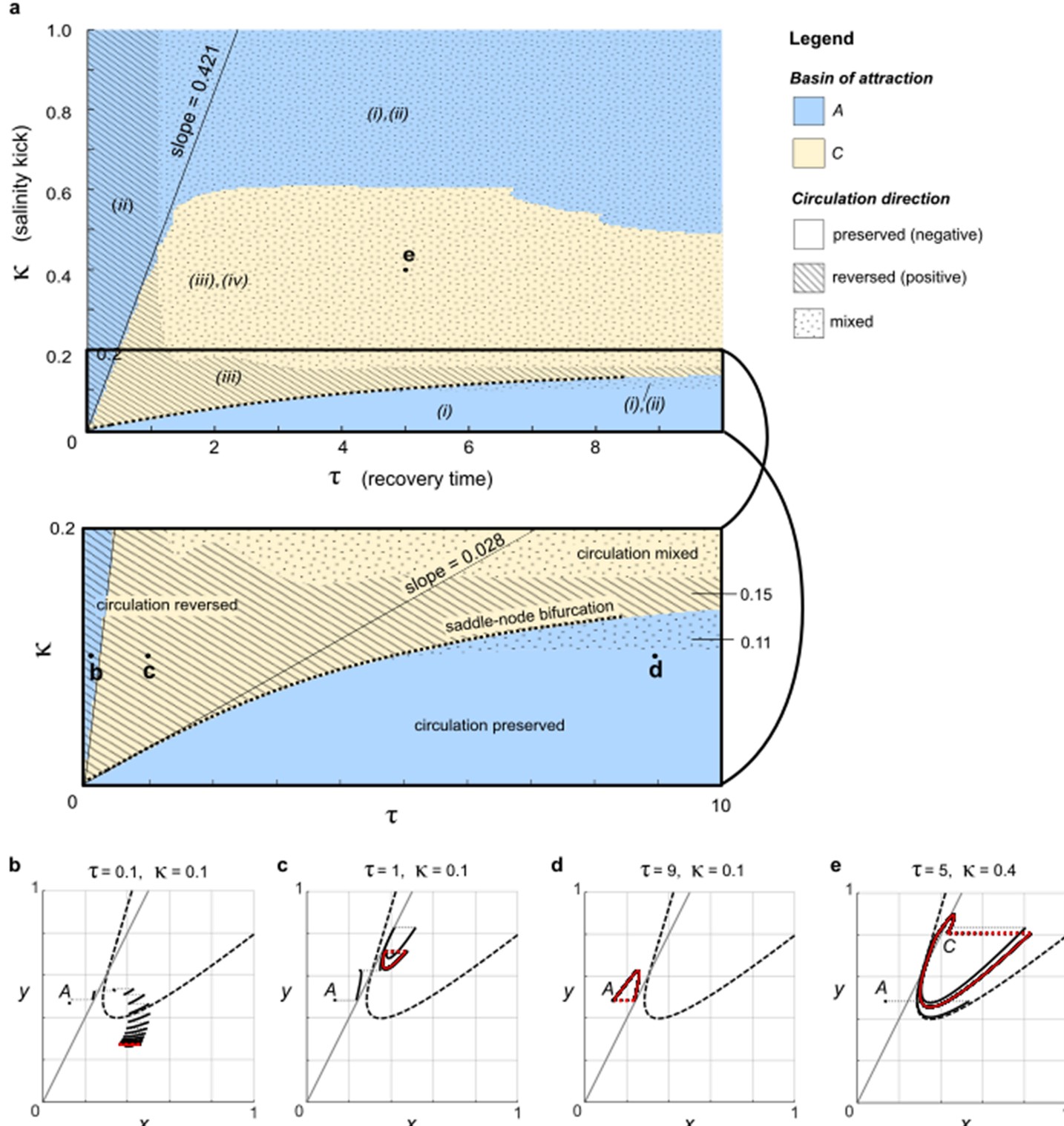

**Fig 3. Basin and circulation outcomes.** Panel (a) categorizes $(\tau, \kappa)$ disturbance space according to the long-term behavior of flow-kick trajectories that start at equilibrium $A$ of Stommel's model (3) and receive kicks $(\kappa, 0)$ every $\tau$ time units. Trajectories can stabilize in the basin of $A$ (blue) or in the basin of $C$ (light yellow). Furthermore, the long-term circulation direction can be reversed (hatched), mixed (dotted) or preserved (no pattern) relative to $A$. Numerals (i)-(iv) designate the region(s) of state space from Fig 2 that the flow-kick trajectory occupies in the long term. Panels (b)–(e) illustrate four qualitatively different combinations of behaviors in state space for the corresponding $(\tau, \kappa)$ parameter combinations shown in panel a.

The blue region (i) in Fig 3a tells another part of the story: when $\kappa < 0.15$, increasing recovery time $\tau$ sufficiently far can cause a trajectory from $A$ to once again stabilize in $A$'s basin. Fig 3d illustrates this possibility for $\tau = 9$ and $\kappa = 0.1$ (disturbance point **d** in Fig 3a). After kicking away from $A$, flowing for time 9 allows the trajectory to almost return to $A$ before the next kick occurs, and a flow-kick fixed point occurs near $A$. When the kick does not surpass the horizontal distance of about 0.15 between $A$ and $A$'s basin boundary (dark blue line in Fig 2), sufficiently long flow times $\tau$ should yield an outcome qualitatively similar to Fig 3d. We therefore expect the lower boundary between basin outcomes in Fig 3a to asymptotically approach the line $\kappa \approx 0.15$. This connection between the lower boundary in disturbance space and a distance to threshold in state space echos similar features of resilience boundaries found in simple flow-kick population models [10, 18].

Stabilizing in the basin of $A$ does not guarantee that ocean circulation direction will be preserved—it only opens the possibility of returning to $A$'s circulation pattern upon cessation of salinity kicks. But in the context of salinity kicks that recur indefinitely, the ultimate direction of ocean circulation is determined not by basin outcome but by the long-term location of a flow-kick trajectory relative to the circulation threshold $y = 2x$. We tested the circulation directions of flow-kick trajectories iterated 100 times from $A$ within the MATLAB script `Fig3a.m`. Despite their different basin fates, both disturbance patterns **b** and **c** fall in the hatched region of disturbance space in Fig 3a where circulation direction reverses in the long term. The only disturbance patterns that preserve the ocean circulation direction in the long term are those such as **d** in the blue, unpatterned region of disturbance space in Fig 3a. It is perhaps not surprising that these correspond to sufficiently small salinity kicks and long recovery times.

Mixed circulation directions over the long-term course of a flow-kick trajectory are also possible, as illustrated in Fig 3e for $(\tau, \kappa) = (5, 0.4)$. More generally, mixed circulation is denoted in Fig 3a with a dotted pattern. This behavior also occurs for large $\tau$ and $0.11 \lesssim \kappa \lesssim 0.15$, because kicks from $A$ in this range cross the circulation threshold but not the basin boundary (see Fig 2). Within Stommel's model, the circulation variable $q(x,y)$ responds instantaneously to salinity $x$ and temperature $y$. We refrain from speculating whether mixed circulation directions could play out in a physical system with inertia.

Despite its complexity, one takeaway from Fig 3 is clear: for a flow-kick trajectory initiated at $A$, only disturbances from the unpatterned region (i) of disturbance space preserve $A$'s circulation direction while stabilizing the trajectory within the basin of $A$. In particular, the short flow times illustrated in Fig 3b that ultimately retain a trajectory within the basin of $A$ still change the model behavior drastically by reversing the circulation direction.

## 4 Continuation and bifurcation of flow-kick fixed points

Although Fig 3 contextualizes recovery-induced tipping in Stommel's kicked model, it does not reveal the cause. Meyer and colleagues [10] attribute the tipping behavior shown in Fig 3c to the fact that longer flow times allow the system to move away from $A$ during the recovery phase, following transient portions of trajectories like the one marked with * in Fig 2. In this section, we will further explain recovery-induced tipping using a bifurcation analysis of flow-kick fixed points in the disturbance parameters $\tau$ and $\kappa$.

When kicks are sufficiently small and recovery times are sufficiently large, the equilibria $A$, $B$, and $C$ of Stommel's undisturbed system continue to analogous flow-kick fixed points $\widehat{A}$, $\widehat{B}$, and $\widehat{C}$. For example, setting $(\tau, \kappa) = (9, 0.1)$, we identified the stable flow-kick fixed points $\widehat{A}$ and $\widehat{C}$ and the unstable flow-kick fixed point $\widehat{B}$ shown in Fig 4a. Flow-kick fixed points were found by applying Newton's method with an error tolerance of $10^{-10}$ to the function

$$\vec{F}(x, y) = \vec{G}_{\tau,\kappa}(x, y) - \begin{bmatrix} x \\ y \end{bmatrix}$$

using the MATLAB script `Fig4a.m`, which calls functions `Newton_2d.m` and `CoupledVar_2d.m`. The latter function computes Jacobian matrices for Newton's method numerically using the variational equation for the spatial derivative of a flow. Stability of flow-kick fixed points was inferred from behavior of nearby flow-kick trajectories.

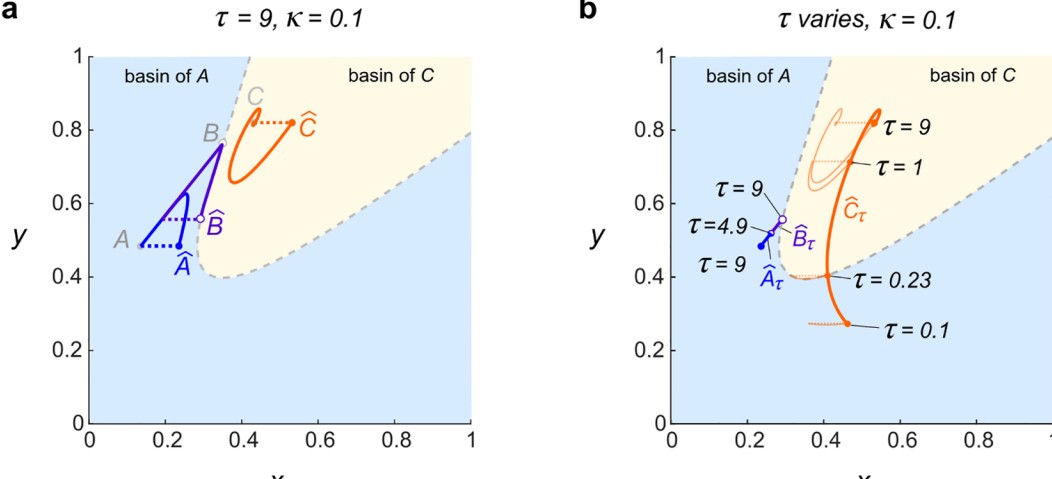

**Fig 4. (a) Three flow-kick fixed points exist for** $(\tau, \kappa) = (9, 0.1)$**: stable fixed points** $\widehat{A}$ **and** $\widehat{C}$ **and unstable fixed point** $\widehat{B}$. (b) As $\tau$ decreases from 9, the branches of fixed points maintain their stability, but $\widehat{A}_\tau$ and $\widehat{B}_\tau$ undergo a saddle-node bifurcation at $\tau = 4.9$ and $(x, y) \approx (0.26, 0.51)$, while the branch $\widehat{C}$ persists and crosses into the original basin of A at $\tau = 0.23$ and $(x, y) \approx (0.41, 0.41)$. Light orange curves illustrate flows and kicks at $\tau = 9$, $\tau = 1$ (compare Fig 3c), $\tau = 0.23$, and $\tau = 0.1$ (compare Fig 3b).

Increasing the kick size $\kappa$ and/or decreasing the flow time $\tau$ can cause fixed points $\widehat{A}$ and $\widehat{B}$ to coalesce and disappear via a saddle-node bifurcation. The dashed curve in Fig 3a traces saddle-node bifurcation values of $(\tau, \kappa)$ and was produced numerically using the MATLAB package MatcontM [19]. We expect that below the dashed curve, the qualitative picture resembles Fig 4a, with three flow-kick fixed points. Above the dashed curve, and for $0 < \kappa \leq 0.4$, numerical simulations using the script `FKGlobalStability.m` suggest the existence of a globally attracting flow-kick fixed point $\widehat{C}$.

The different basin outcomes above the dashed saddle-node bifurcation curve in Fig 3a stem from different positions of the remaining attracting flow-kick equilibrium or equilibria. Fig 4b illustrates how $\widehat{A}$, $\widehat{B}$, and $\widehat{C}$ move as $\tau$ varies and $\kappa = 0.1$ remains constant. This numerical continuation was carried out in MATLAB using the script `Fig4b.m`, which also calls functions `Newton_2d.m` and `CoupledVar_2d.m`. For $\kappa = 0.1$, the saddle-node bifurcation occurs at $\tau \approx 4.9$. When $0.23 < \tau < 4.9$, the globally attracting fixed point $\widehat{C}$ lies in the basin of C and flow-kick trajectories from A tip into the basin of C as shown in Fig 3c. For even more frequent disturbances ($0.1 \leq \tau < 0.23$), the globally attracting fixed point $\widehat{C}$ moves into the basin of A and flow-kick trajectories from A stabilize in the basin of A as shown in Fig 3b. When we reverse the order of these events and consider $\tau$ increasing from 0.1 to 1, we see that the passage of $\widehat{C}$ from the basin of A to the basin of C is responsible for the recovery-induced tipping phenomenon. See Sect 6.1 for discussion of the broader relevance of recovery-induced tipping to resilience in dynamical systems.

## 5 Continuous salinity disturbances

Kicks are an idealized representation of freshwater inputs to the North Atlantic, which appear in datasets as continuous but peaked discharge curves (e.g. [17,20]). Instead of using discrete kicks to model salinity disturbances, others have incorporated a constant "virtual salt flux" [14] or time-varying salinity forcing [21] into ODE models of thermohaline circulation. In Stommel's nondimensional model (3), one might embed a continuous salinity perturbation $r$ as follows:

$$x' = \delta(1 - x) - \frac{1}{\lambda}|Rx - y|x + r \tag{5a}$$

$$y' = 1 - y - \frac{1}{\lambda}|Rx - y|y. \tag{5b}$$

 

Interpreting *r* as an average rate of salinity perturbation, we explore the extent to which the continuous disturbance model (5) reproduces the dynamics of the more complex flow-kick model defined by Eqs (3) and (4).

To compare across models, we identify each constant disturbance rate $r \geq 0$ with a family of flow-kick maps $\tau, \kappa$ for which the average disturbance rate $\kappa/\tau$ matches *r*. These parameter combinations form a line of slope *r* through the origin in $(\tau, \kappa)$ disturbance space. Matching flow-kick and ODE models by average disturbance rate is more than a heuristic choice: it is known that in the limit as $\tau$ and $\kappa$ approach zero along a ray with $\kappa/\tau = r$ fixed, the flow-kick map $\tau, \kappa$ generates the vector field given by Eqs (5) [10,11].

Previous work indicates that features such as hyperbolic equilibria and saddle-node bifurcations in the ODE system (5) continue to flow-kick systems with matching disturbance rates when the flow time is sufficiently small [11]. However, flow-kick models are also known to generate novel dynamics relative to their continuous analogs (e.g. [6]). Determining whether the continuously perturbed Stommel model replicates behaviors of the kicked model (3) and (4) thus requires analysis of the particular continuous system (5).

### 5.1 Bifurcation in *r*

Changes to the salinity kick size $\kappa$ or recovery time $\tau$ both translate to changes in the continuous salinity perturbation parameter $r \equiv \kappa/\tau$. Fig 5a presents a bifurcation diagram in the salinity disturbance rate *r* for the continuous system (5), generated using the MATLAB script `Fig5.m`. The vertical axis represents the *y* coordinate of equilibria. At $r = 0$ we recognize the equilibria *A*, *B*, and *C* of Stommel's model. At $r \approx 0.028$, the branches of equilibria emanating from *A* and *B* undergo a saddle-node bifurcation, eliminating bistability from the system. Numerical simulations indicate that once *r* exceeds $\approx 0.028$, the branch $C_r$ of equilibria emanating from *C* is not only locally stable but attracts all trajectories within $[0,1] \times [0,1]$ (see `CtsGlobalStability.m`).

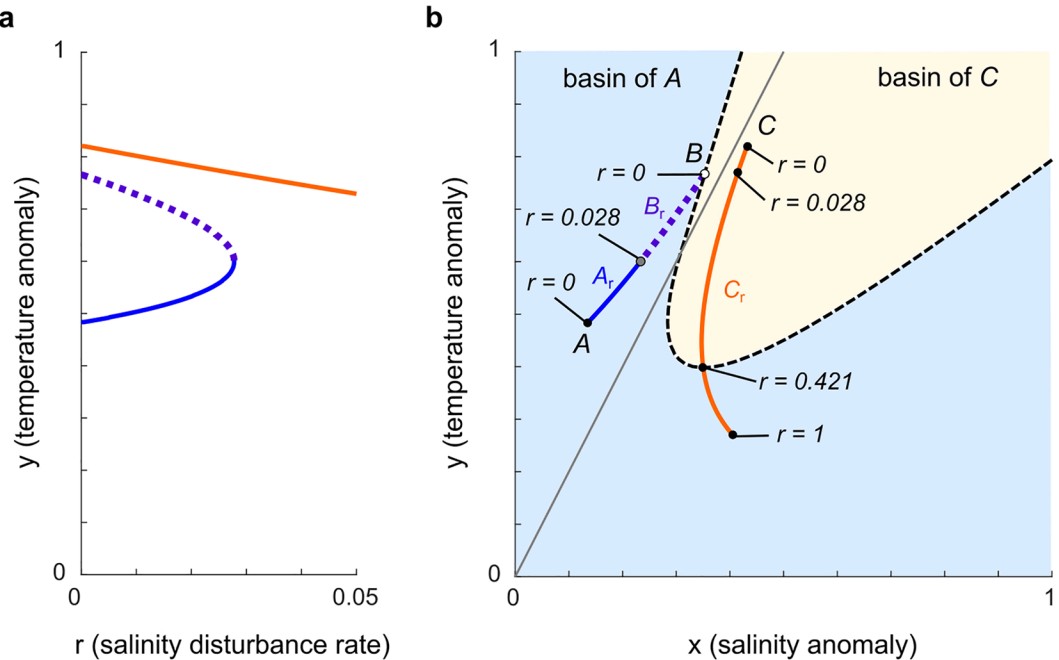

**Fig 5**. **Bifurcation diagram showing the transition of equilibria as disturbance ratio (*r*) increases.** A saddle-node bifurcation at $r \approx 0.028$ marks the collapse from three equilibria to a single equilibrium. The vertical axis represents the *y* coordinate of the equilibria.

Examining the line with slope $\kappa/\tau = 0.028$ in Fig 3, we see that the $r$-value at the saddle-node bifurcation in the continuous model (5) closely predicts the location of the saddle-node bifurcation in the flow-kick model (3) and (4) for small recovery times ($0 \leq \tau \lesssim 2$). In this sense, the continuous model captures a key bifurcation in the flow-kick model. But as recovery times increase, the bifurcation threshold for the flow-kick system departs from the rate $\kappa/\tau = r = 0.028$. This departure, which also occurs between continuous and flow-kick models of nutrient inputs to freshwater lakes [10] and rainfall inputs to semi-arid ecosystems [11], points to the value of flow-kick models for assessing tipping thresholds in systems with truly discrete disturbance regimes.

## 5.2 Paths of equilibria

In Fig 5b, we demonstrate an analog to recovery-induced tipping in the continuously perturbed model (5) by plotting branches of equilibria parametrically as functions of $r$. Numerical continuation was performed in MATLAB using the script `Fig5.m`, which calls the function `Newton_cts.m`. We denote the branches of equilibria as $A_r$ in blue, $B_r$ in purple, and $C_r$ in orange (with $A_0 = A$, $B_0 = B$, and $C_0 = C$). As $r$ increases from 0 beyond 0.028, $A_r$ and $B_r$ converge and undergo a saddle-node bifurcation, while $C_r$ persists along a curvilinear path.

Crucially, the branch $C_r$ crosses the separatrix (dashed curve) between the original basins of $A$ and $C$ at $r \approx 0.421$. This crossing explains the system's counterintuitive behavior: if a trajectory starts near $A$ and $r$ exceeds the bifurcation value of about 0.028, then the trajectory must converge toward $C_r$, and the position of $C_r$ relative to the separatrix determines the basin fate of a disturbed trajectory.

Translating flow-kick disturbance regimes to the continuous model with $r = \kappa/\tau$, the frequent disturbance regime $(\tau, \kappa) = (0.1, 0.1)$ shown in Fig 3b corresponds to an average salinity disturbance rate of $r = 1$, which puts $C_1$ in the original basin of $A$. A solution to the ODE (5) with $r = 1$ that starts from $A$ tends towards $C_1$ and winds up in the basin of $A$, just as the flow-kick trajectory in Fig 3b stabilizes in the basin of $A$. On the other hand, the less frequent disturbance regime $(\tau, \kappa) = (1, 0.1)$ shown in Fig 3c corresponds to an average salinity disturbance rate of $r = 0.1$, which puts $C_1$ in the original basin of $C$. For $r = 0.1$, a trajectory from $A$ stabilizes within the basin of $C$. In this way the continuous model replicates the phenomenon that increasing the recovery time between salinity kicks (thereby decreasing the average salinity disturbance rate) can trigger tipping into the basin of $C$.

The critical disturbance rate $r = 0.421$ that separates basin outcomes in the continuous model is shown in Fig 3a via the line through the origin with slope $\kappa/\tau = 0.421$. The continuous model does a nice job at predicting the boundary between flow-kick basin outcomes for $0 < \kappa \lesssim 0.4$ and $0 < \tau \lesssim 1$.

In summary, the continuous model replicates two key features of the flow-kick model: first, a saddle-node bifurcation destroys the temperature-dominated equilibrium $A$, and second, the equilibrium $C$ moves into $A$'s original basin as disturbance rates increase, generating a phenomenon analogous to the recovery-induced tipping that was originally reported by [10] in the flow-kick model.

# 6 Discussion

## 6.1 Dynamic implications of recovery-induced tipping

The recovery-induced tipping detected in Stommel's model points to surprising possibilities within dynamic models of disturbance and resilience. We have seen in both the discrete (4) and continuous (5) models of salinity perturbation that increasing the rate of disturbance causes an alternative equilibrium $C$ to become globally attracting and ultimately shift into the original basin of an attracting equilibrium $A$ for the undisturbed system. Provided that the high disturbance rate ($r = \frac{\kappa}{\tau} \gtrsim 0.421$) persists long enough for a trajectory to converge to the shifted equilibrium, the trajectory will flow back to $A$ when the disturbance ceases. In contrast, perturbing salinity at a moderate rate ($0.028 \lesssim r = \frac{\kappa}{\tau} \lesssim 0.0421$) causes a trajectory to equilibrate in the basin of $C$, where it remains stuck upon cessation of disturbance. This suggests that increasing a disturbance rate can offer an escape from hysteresis. Fig 6 illustrates this essential mechanism and its implications in

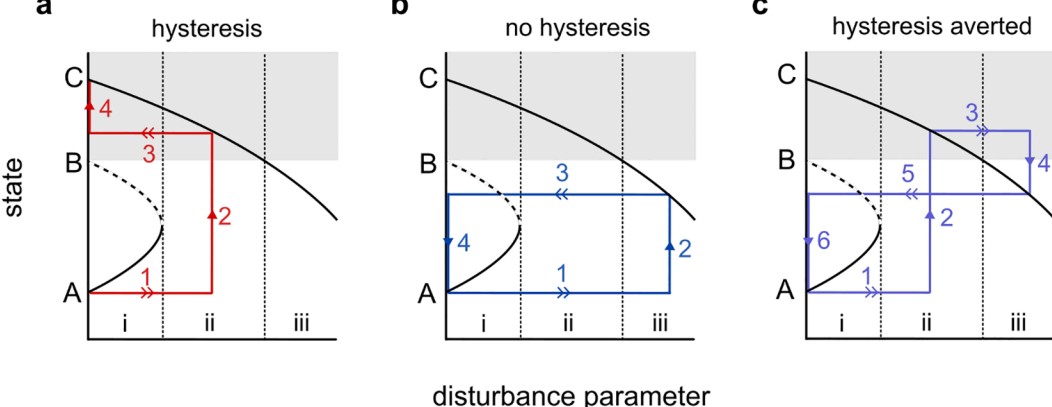

**Fig 6. Bifurcation diagram for a hypothetical system.** Intensifying disturbance first destroys an attracting fixed point *A* via a saddle-node bifurcation (i→ii) and then shifts an alternative attractor *C* into *A*'s original basin (ii→iii). Temporary, moderate disturbance can cause hysteresis (panel a), while temporary, high-intensity disturbance can avoid hysteresis (panel b). Further, increasing disturbance intensity from moderate to high can avert hysteresis (panel c). Double arrows indicate rapid parameter changes while single arrows indicate evolution of state. Arabic numerals indicate event order.

a hypothetical bifurcation diagram. We expect this behavior more generally in systems in which intensifying disturbance eliminates one attractor and drives an alternative attractor into its original basin. Intensifying and then ceasing a disturbance to ultimately undo its effects would be a counterintuitive strategy for managing an ecosystem or climate subsystem. But perhaps this strategy may be relevant in some situations. While these authors do not endorse large-scale experimentation, the phenomenon certainly warrants further study.

### 6.2 Model comparison to geophysical measurements

In addition to viewing recovery-induced tipping as an abstract phenomenon, one can ask how the salinity kicks and recovery times that give rise to recovery-induced tipping in Stommel's model compare to actual salinity disturbances in the Atlantic Ocean. Table 1 summarizes flow-kick parameters estimated for two disturbance sources: Great Salinity Anomalies (GSAs) and annual Greenland melt, discussed below. The Subpolar North Atlantic experienced GSAs in the 1970s, 1980s, and 1990s [22], with measured salinity variability on the order of 0.1 to 0.5 practical salinity units (PSU) [23, 24]. These recurrent salinity drops are attributed to influx of ice and freshwater from the Arctic Ocean [23]. Again starting around 2015, salinity dropped by up to 0.4 PSU in the top 100 meters of the Iceland basin [25] and by 0.1 PSU in the top 200 meters of the Irminger Sea [23], a key site of North Atlantic Deep Water formation. We estimate that these drops in salinity correspond to changes in the range of +0.02 to +0.1 to Stommel's nondimensional salinity variable $x$ (see S2 File). Thus in the context of GSAs, the salinity kicks of $\kappa = 0.1$ studied here and by [10] appear large but not impossible.

On the other hand, using a temperature relaxation timescale of 25 days [21], we estimate that time periods between Great Salinity Anomalies reported by [23] correspond to flow times $100 \leq \tau \leq 320$ (see S2 File). Such long recovery periods, which are two orders of magnitude larger than the flow times for which recovery-induced tipping occurs in Stommel's

**Table 1.** Flow-kick parameter estimates for two geophysical processes that perturb North Atlantic salinity.

| Salinity perturbation source | Estimated kicks | Estimated flow times |
|---|---|---|
| Great Salinity Anomalies | $0.02 \leq \kappa \leq 0.1$ | $100 < \tau < 320$ |
| Annual Greenland melt | $0.007 \leq \kappa \leq 0.04$ | $\tau \approx 15$ |
| Methods of parameter estimation are described in Sect 6.2 and S2 File. | | |

kicked model, allow the model system to converge near an attracting equilibrium $A$ or $C$ after each kick. A GSA-like disturbance regime with $100 < \tau < 320$ and $0.02 \leq \kappa \leq 0.1$ falls in region (i) of Fig 3a. Here, trajectories that start near $A$ preserve the overall temperature-dominated circulation regime—as shown in Fig 3d—though circulation does weaken periodically with each kick before the system converges back towards $A$. Although any quantitative comparisons between Stommel's simple model and real-world data should be treated with caution, we note that a weakening temperature-dominated circulation regime in Stommel's model is consistent with modern observations of the AMOC [26].

Glacial meltwater is another mechanism behind salinity perturbations in the North Atlantic. During the last deglaciation period, meltwater from the Laurentide Ice Sheet is thought to have decreased surface salinity and density enough to halt the production of North Atlantic Deep Water and shut down the AMOC [27,28]. In modern times, Greenland is a significant source of freshwater for the North Atlantic, with influx rates peaking annually in the summer months [17]. During the decade 2000–2010, Greenland contributed an excess freshwater flux of approximately 100 km$^3$ yr$^{-1}$ to the Irminger Sea relative to baseline flux during 1960–1980 and an absolute flux around 350 km$^3$ yr$^{-1}$ (S1 Fig in [17]). This freshwater flux includes glacial meltwater, tundra runoff, and solid ice discharge. Freshwater additions of 100–350 km$^3$, if instantaneous and well-mixed in the surface waters of the Irminger Sea, translate to nondimensional salinity kicks $+0.007 \lesssim \kappa \lesssim +0.04$, to one significant figure (see S2 File). A kick of 0.1 to the salinity variable in Stommel's model could be achieved locally in the Irminger Sea if roughly 3–14 years of freshwater input from Greenland arrived all at once. Furthermore, an annual return period translates to a flow time $\tau \approx 15$ (see S2 File). Thus a $(\tau, \kappa)$ disturbance regime modeled after Greenland inputs again lies in region (i) of Fig 3a, and is likely far away from any recovery-induced tipping threshold.

### 6.3 Future directions

The recovery-induced tipping that we have investigated within Stommel's two-box model suggests future directions both in dynamic models of ocean circulation and in the mathematical theory of tipping points. In the context of ocean circulation models, one might explore whether recovery-induced tipping behavior persists when stochasticity is incorporated into flow times and/or kick sizes. It would also be interesting to determine whether the phenomenon can occur in other models of ocean circulation, either conceptual (e.g. [21]) or higher complexity (e.g. [29]).

From a theoretical standpoint, one might explore which conditions are necessary and/or sufficient to produce the dynamic behavior illustrated in the bifurcation diagrams of Figs 4b, 5b, and 6: namely, an alternative stable state becoming globally stable and migrating into the original basin of attraction of a previously stable state as a disturbance pattern intensifies. Determining the factors that produce this behavior would improve our understanding of when to expect recovery-induced tipping and the related phenomenon that intensifying disturbance can provide an escape from hysteresis.

### Supporting information

**S1 File. Stommel model reduction.** This document supplies further details on the reduction from Eq (1) to Eq (3). (PDF)

**S2 File. Estimating kicks and flow times from geophysical data.** This document presents rough estimates of kicks and flow times corresponding to Great Salinity Anomalies in the North Atlantic and annual freshwater input from Greenland. (PDF)

### Acknowledgments

We thank Alanna Hoyer-Leitzel and Sarah Iams for sharing code that adapts MatcontM for flow-kick maps; this provided a starting point for tracing out the saddle-node bifurcation curve in Fig 3a. Carleton graduates Horace Fusco and Collin Smith developed the code `Newton_2d.m` that we used to find flow-kick fixed points.

## Author contributions

**Conceptualization:** Katherine Meyer.

**Data curation:** Peter Kelly, Xinyi Leng.

**Formal analysis:** Peter Kelly, Xinyi Leng, Albert Jing, Katherine Meyer.

**Funding acquisition:** Katherine Meyer.

**Investigation:** Peter Kelly, Xinyi Leng, Pascal Cogan, Albert Jing, Katherine Meyer.

**Methodology:** Peter Kelly, Katherine Meyer.

**Project administration:** Katherine Meyer.

**Software:** Peter Kelly, Katherine Meyer.

**Supervision:** Katherine Meyer.

**Validation:** Peter Kelly, Pascal Cogan, Albert Jing.

**Visualization:** Peter Kelly, Xinyi Leng, Pascal Cogan, Albert Jing, Katherine Meyer.

**Writing – original draft:** Katherine Meyer.

**Writing – review & editing:** Peter Kelly, Xinyi Leng, Albert Jing, Katherine Meyer.

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
