## [Decision Letter · Decision Letter 0]

30 Jul 2025

PONE-D-25-36894

Recovery-induced tipping in Stommel's kicked ocean box model

PLOS ONE

Dear Dr. Meyer,

Thank you for submitting your manuscript to PLOS ONE. After careful consideration, we feel that it has merit but does not fully meet PLOS ONE’s publication criteria as it currently stands. Therefore, we invite you to submit a revised version of the manuscript that addresses the points raised during the review process.

We look forward to receiving your revised manuscript.

Kind regards,

Sandipan Mondal, Ph.D.

Academic Editor

PLOS ONE

Journal Requirements:

2. We notice that your supplementary [figures/tables] are included in the manuscript file. Please remove them and upload them with the file type 'Supporting Information'. Please ensure that each Supporting Information file has a legend listed in the manuscript after the references list.

Reviewers' comments:

Reviewer's Responses to Questions

**Comments to the Author**

1. Is the manuscript technically sound, and do the data support the conclusions?

Reviewer #1: Yes

Reviewer #2: Yes

2. Has the statistical analysis been performed appropriately and rigorously?

Reviewer #1: Yes

Reviewer #2: Yes

3. Have the authors made all data underlying the findings in their manuscript fully available?

Reviewer #1: Yes

Reviewer #2: No

4. Is the manuscript presented in an intelligible fashion and written in standard English?

Reviewer #1: Yes

Reviewer #2: Yes

5. Review Comments to the Author

Reviewer #1: I have read the manuscript titled “Recovery-induced tipping in Stommel’s kicked ocean box model” with great interest. Although I am not a specialist in dynamical systems or theoretical oceanography, I found the study intellectually engaging and educational. The clarity of exposition and the elegant connection between nonlinear system dynamics and real-world climate perturbations allowed me to follow the arguments and learn from the methodology and interpretations presented. I commend the authors for producing a manuscript that is accessible beyond their immediate discipline. For the rest, please find the attached comments in the PDF file attached.

Reviewer #2: The manuscript is well written and easy to understand. However, the authors must improve the quality of the plots. The authors must also include recent citations in the introduction part of the manuscript.

6. PLOS authors have the option to publish the peer review history of their article (what does this mean?). If published, this will include your full peer review and any attached files.

Reviewer #1: **Yes:** Aratrika Ray

Reviewer #2: No

---

## [Author Response · Author response to Decision Letter 1]

7 Jan 2026

Our responses to specific reviewer comments can be found in the file "Response to Reviewers" uploaded in the Attach Files stage.

---

## [Editor Report · Decision Letter 1]

19 Jan 2026

Recovery-induced tipping in Stommel's kicked ocean box model

PONE-D-25-36894R1

Dear Dr. Katherine,

We’re pleased to inform you that your manuscript has been judged scientifically suitable for publication and will be formally accepted for publication once it meets all outstanding technical requirements.

Kind regards,

Sandipan Mondal, Ph.D.

Academic Editor

PLOS One
---

## [Editor Report · Acceptance letter]

PONE-D-25-36894R1

PLOS One

Dear Dr. Meyer,

I'm pleased to inform you that your manuscript has been deemed suitable for publication in PLOS One. Congratulations! Your manuscript is now being handed over to our production team.

Kind regards,

on behalf of

Dr. Sandipan Mondal

Academic Editor

PLOS One